# Feasibility of a new electronic patient-reported outcome (ePRO) system for an advanced therapy clinical trial in immune-mediated inflammatory disease (PROmics): protocol for a qualitative feasibility study

Sarah E Hughes [1,2,3,4] Christel McMullan [1,4,5,6] Anna Rowe,[7,8] Ameeta Retzer [1,3,5] Rebecca Malpass,[7,8] Camilla Bathurst,[7,8] Elin Haf Davies,[9] Chris Frost,[9] Gary McNamara,[10] Rosie Harding,[11] Gary Price,[1] Roger Wilson [12] Anita Walker,[1] Philip N Newsome,[1,7,13,14] Melanie Calvert [1,2,3,4,5,7,14,15]

For numbered affiliations see end of article.

**Correspondence to**
Dr Sarah E Hughes;
s.e.hughes@bham.ac.uk

## ABSTRACT

**Introduction** The use of electronic patient-reported outcome (ePRO) systems to capture PRO data in clinical trials is increasing; however, their feasibility, acceptability and utility in clinical trials of advanced therapy medicinal products (ATMPs) are not yet well understood. This protocol describes a qualitative study that aims to evaluate the feasibility and acceptability of ePRO data capture using a trial-specific ePRO system (the PROmics system) within an advanced therapy trial involving patients with immune-mediated inflammatory disease (rheumatoid arthritis, lupus, primary sclerosing cholangitis (PSC) and Crohn's disease).

**Methods and analysis** This protocol for a remote, qualitative, interview-based feasibility study is embedded within the POLARISE trial, a single-arm, phase II, multisite ATMP basket trial in the UK. 10–15 patients enrolled in the POLARISE trial and 10–15 research team members at the trial sites will be recruited. Participants will take part in semistructured interviews which will be transcribed verbatim and analysed thematically according to the framework method. Data collection and analysis will occur concurrently and iteratively. Researcher triangulation will be used to achieve a consensus-based analysis, enhancing rigour and trustworthiness.

**Ethics and dissemination** This study was approved by the London—West London and GTAC Research Ethics Committee (Ref: 21/LO/0475). Informed consent will be obtained from all participants prior to data collection. The study findings will be published in peer-review journals and disseminated via conference presentations and other media. Our patient and public involvement and engagement group and ATMP stakeholder networks will be consulted to maximise dissemination and impact.

**Trial registration number** ISRCTN80103507.

## INTRODUCTION

The use of stem cells, gene therapy and tissue engineering in the treatment of disease or injury represents an emerging and rapidly developing approach to medicine.[1 2] These advanced therapy medicinal products (ATMPs) have the potential to transform the treatment of some diseases (eg, sickle cell disease, cancer and haemophilia), removing the need for long-term medical care. In the UK, there has been steady year-on-year growth in the number of clinical trials of ATMPs.[3]

Endpoints in clinical trials are designed to assess the effects of an intervention, including the efficacy, safety and tolerability of the medicinal product. As an adjunct to

---

**STRENGTHS AND LIMITATIONS OF THIS STUDY**

⇒ The use of an electronic patient-reported outcome (ePRO) system to collect PRO data as part of an early-phase ATMP clinical trial with patients with immune-mediated inflammatory disease could help to identify symptom burden and adverse side effects in research participants earlier, enhancing patient safety.

⇒ This study will provide data on the feasibility and acceptability of the PROmics approach to enhance future PRO strategy in later phases, promoting efficient use of measures; data completeness; acceptability to trial participants; and inform future analyses and sample size estimation.

⇒ The study sample will include research participants from National Health Service (NHS) hospitals from different regions of the UK, which will enhance the transferability and generalisability of the study findings.

⇒ The study sample will include only trial participants who consent to take part in this ePRO feasibility substudy; therefore, recruitment may be subject to selection bias.

BMJ

clinical endpoint data, patient-reported outcome (PRO) data can provide further, valuable evidence of treatment benefit or risk.[4 5] A PRO is 'any report of the status of a patient's health condition that comes directly from the patient, without interpretation of the patient's response by a clinician or anyone else'.[5] PRO data include patient perspectives of symptom burden, functioning and their health-related quality of life (QoL) typically captured using validated, self-report questionnaires. PROs may be used alone or in combination with wearable devices to collect real time, objective measures of patient activity. In addition to their use as trial endpoints, PRO data may be used to support pharmaceutical labelling claims, clinical decision-making, clinical guideline development and inform health policy.[6]

PROs have been traditionally assessed using paper-based questionnaires that have been evaluated psychometrically to establish their measurement properties. Increasingly PRO instruments are administered digitally as electronic PROs (ePROs). ePRO systems typically use a digital device such as a computer, tablet or smartphone to deliver the questionnaires to the patient.[7] Leveraging remote connections and notification capability, ePROs can be programmed to trigger alerts to the clinical/trials staff should a participant report the presence of one or more symptoms of clinical concern.[8 9] In addition, ePROs offer the advantage of reducing data entry errors and respondent and administrative burden while facilitating data collection at specified time points with a record of the date and time entries are made.[7] Remote monitoring functionality enables trial participants to remain in their own homes at the same time as facilitating the surveillance of symptoms and disease status by the trial's clinical team.[8 10]

### PROs in trials of ATMPs

Due to their novelty, complexity and technical specificity, ATMPs bring 'new, unexplored risks to patients', including risks such as unwanted immunogenicity and severe toxicity.[11] PROs may be used to directly capture symptomatic adverse events (AEs) and the overall treatment burden from the patient perspective and, therefore, have potential to provide evidence on the safety and efficacy of an ATMP under investigation. In this way, PRO data can support post-treatment follow-up and risk management (including monitoring of the longer term effects on patients), provide essential data to inform real-world use, facilitate cost-benefit profiling, as well as support marketing authorisation.[2 12] In addition, PROs have potential to support a fuller understanding of tolerability (the ability or desire of the patient to adhere to a specific dose or intensity of therapy) by providing direct measurements from the patient on how they are feeling and functioning while on treatment.[13 14] For patients, PROs in ATMP trials provide an opportunity to communicate outcomes of importance not captured by traditional clinical endpoints. PROs may also encourage patients to engage as participants, increase the likelihood of PRO

claims in product labelling, and empower patients and clinicians to make more informed treatment decisions leading to better clinical outcomes.[15]

### The PROmics ePRO system

The PROmics system is a trial-specific electronic data capture system designed to collect and assess PRO data when patients receive an advanced cell therapy. PROmics captures PRO data relating to patient-reported AEs and side effects, global tolerability of the medicinal product, QoL and daily functioning tailored for each disease cohort. The PRO instruments selected for PROmics were identified in collaboration with patient partners and other stakeholders in a workshop held in 2018. The included PROs were selected to facilitate assessment of efficacy and tolerability by gathering preliminary evidence from the patient perspective on the benefits and risks of the ATMP. These data also have potential to inform the design and conduct of later-phase trials.[16] In addition to patient preference, selection was based on regulatory recommendations, the psychometric properties of the PRO instruments, use in clinical practice and trials, and a need to ensure consistency in PRO reporting across disease cohorts (table 1).

The following PRO instruments were selected for inclusion:
- ► The National Cancer Institute PRO Version of the Common Terminology Criteria for Adverse Events (PRO-CTCAE) is a 78-item questionnaire that has been developed to characterise the frequency, severity and interference of symptomatic treatment toxicities from the patient perspective. The PRO-CTCAE was designed to be used as a companion to the Common Terminology Criteria for Adverse Events (CTCAE), the standard lexicon for AE reporting in trials.[17–19]
- ► The FACT-G GP5 is a single item measuring patient-reported global treatment tolerability.[20]
- ► The EuroQol 5 Dimensions-5 Level (EQ-5D-5L) is a generic measure of self-rated health status. It consists of five items, each assessing a different health domain (mobility, self-care, usual activities, pain/discomfort, anxiety/depression), and a Visual Analogue Scale (VAS) measuring global health status.[21]
- ► The FACIT-Fatigue scale is a 13-item generic PRO instrument measuring QoL concerns relating to fatigue in chronic illness.[22 23]
- ► The Fatigue Severity Scale is a nine-item generic instrument that assesses the severity of fatigue and its impact on daily living.[24]
- ► The Primary Sclerosing Cholangitis PRO (PSC-PRO) instrument is a 42-item questionnaire measuring PSC symptoms and their impact.[25]

To complete the PRO instruments, patients access the PROmics system via an app on their personal tablet or smartphone, a data collection system known as 'Bring Your Own Device (BYOD)'.[7] The PRO instruments are delivered in a standardised order for self-completion by the trial participant. Preprogrammed notifications will

**Table 1** PRO instruments included in the PROmics system for the POLARISE trial

| PRO instrument | Concept of interest | Disease cohort | | | | Rationale for selection |
| | | PSC | RA | LN | CD | |
| --- | --- | --- | --- | --- | --- | --- |
| PRO-CTCAE | Burden (frequency, severity, interference) of symptomatic toxicities | X | X | X | X | Side effect and adverse event monitoring |
| FACT-G GP5 | Global tolerability of treatment | X | X | X | X | Monitoring of treatment tolerability from the patient perspective |
| EuroQol 5 Dimensions-5 Level | Health status | X | X | X | X | Local health reimbursement requirement |
| Fatigue Severity Scale | Fatigue severity and impact on daily living | X | X | X | X | Preferred by PSC patients, use in previous trials, methodological consistency and to maximise comparability between trial arms |
| FACIT-Fatigue | Quality of life rating to fatigue | X | X | X | X | Recommended for use in RA patients by the European Medicines Association and the International Consortium for Health Outcomes Measurement, methodological consistency, maximise comparability between trial arms |
| PSC-PRO | PSC symptom severity and impact | X | | | | PSC-specific PRO instrument, patient preference, instrument developed with extensive patient input |

CD, Crohn's disease; LN, lupus nephritis; PRO-CTCAE, patient-reported outcomes Version of the Common Terminology Criteria for Adverse Events; PSC, primary sclerosing cholangitis; RA, rheumatoid arthritis.

remind participants to submit their completed questionnaires, if they have not done so within a specified time period. Clinical alerts, in the form of email notifications to the research site team, are triggered if a participant reports a high symptom burden. The alert, in the form of an email message to named trial staff at the relevant research site, prompts staff to login to the PROmics clinical dashboard, review the PRO scores for the corresponding trial ID number, and, if required, complete an AE report using the CTCAE.[26]

## POLARISE trial

The PROmics feasibility study is set in the context of the POLARISE trial (EudraCT: 2019-003404-13; ISRCTN80103507), a single-arm, multicentre, phase II basket trial investigating the safety and activity of ORBCEL-C in the treatment of patients (N=60) with immune-mediated inflammatory disease, specifically, PSC, rheumatoid arthritis, lupus nephritis and Crohn's disease. These disease groups represent a highly heterogeneous sample in terms of age, physical impairment, and other disease-related characteristics. However, all have a shared pathogenesis and are associated with high symptom burden and reduced QoL, necessitating the collection of PROs at scheduled time points as well as notifications of patient-reported AEs.[27] Remote collection of PRO data is possible through use of an ePRO system, with its specific mode of delivery and enhanced functionality, enabling these trial specific requirements to be met.

## Aims and objectives

This study aims to qualitatively assess the feasibility and acceptability of the PROmics system, an ePRO platform for use in trials of ATMPs. The planned study will evaluate the use of PROmics when deployed in a trial environment generally and, more specifically, in the POLARISE trial, a phase II multidisease ATMP trial in patients with immune-mediated inflammatory disease.

The specific study objectives are:
1. To explore with PROmics users, their' experiences of being trained with and using the PROmics ePRO system.
2. To explore decision-making with POLARISE trial participants who did not consent to reporting their PROs via PROmics or withdrew from using the PROmics system.
3. To explore the benefits and shortcomings of the PROmics system and how these may be developed further.
4. To explore how use of the PROmics system was managed by research staff and research participants in the context of the POLARISE trial.

## METHODS AND ANALYSIS

This study will be reported in accordance with the Consolidated Criteria for Reporting Qualitative Research (COREQ).[28]

### Design

A remote, qualitative, interview-based feasibility study (see figure 1) embedded within the POLARISE trial, a

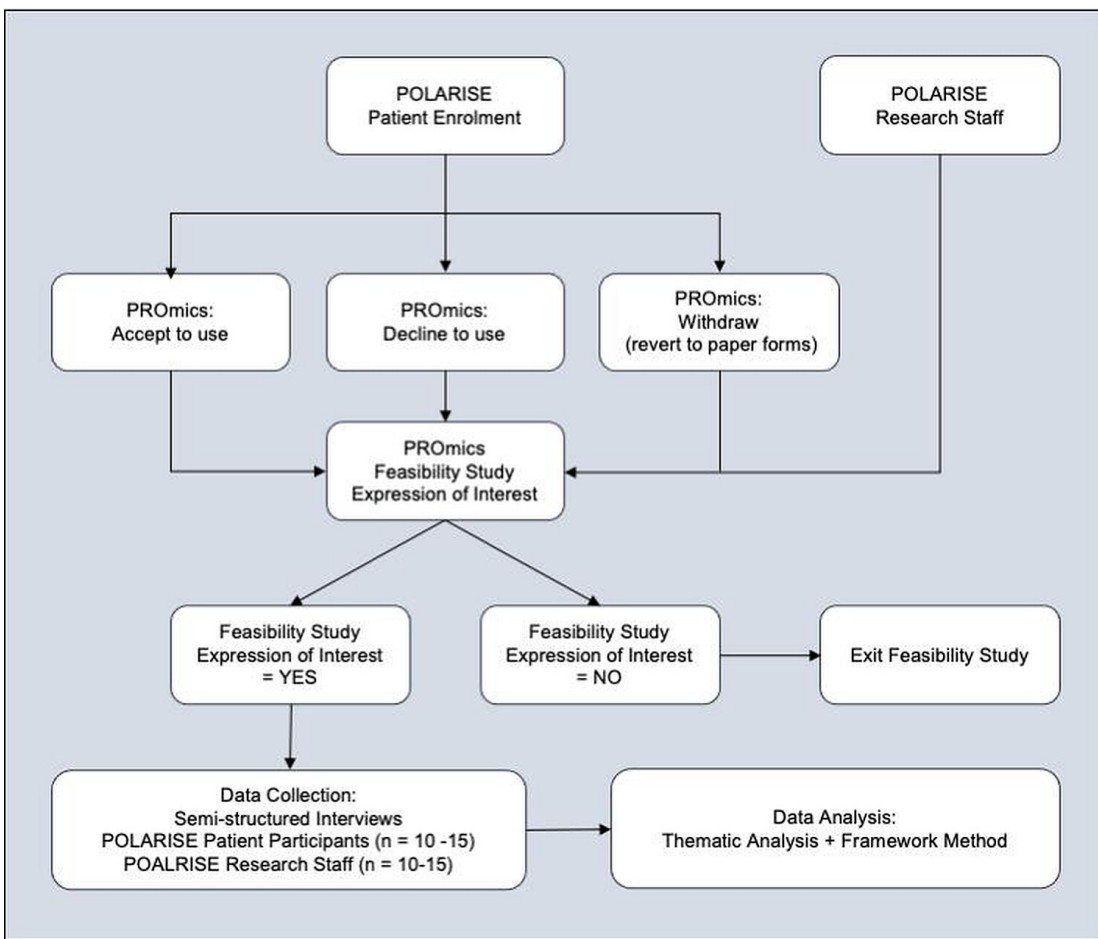

**Figure 1** PROmics qualitative feasibility study embedded within the POLARISE trial—participant flow chart.

single-arm, phase II ATMP basket trial recruiting from six National Health Service (NHS) trusts in the UK.

## Participant selection

In total, it is estimated that 20–30 participants will be recruited to this feasibility study. Recruitment will proceed until data saturation, the point in data collection when new data are redundant, is achieved.[29] Two participant groups, POLARISE trial participants (n=10–15 participants) and members of the POLARISE research team (n=10–15 participants), will be enrolled.

Sampling will be purposive for maximum variation for both POLARISE participants and research staff. Maximum variation where possible will be sought to ensure broad representation of disease groups, age, sex, ethnicity, research site/geographical location for POLARISE participants and age, sex, clinical/research role, research site/geographical location for POLARISE research staff.

To take part in the PROmics feasibility study, patients must be enrolled in the POLARISE trial (therefore meeting the trial eligibility criteria) and have either: (1) consented to report their PRO data (symptoms, tolerability, QoL) using the PROmics ePRO system; (2) consented initially to report their PROs using the PROmics and subsequently withdrew from using the

PROmics system to report their PRO data, reverting to paper questionnaires or (3) declined to report their PROs using the PROmics system. Sampling will aim to recruit between three and five participants from each disease group and any POLARISE participants who decline to use or subsequently withdraw from using the PROmics system.

Participants who are research staff will be members of the research team who have been involved in the planning, delivery, administration and use of the PROmics ePRO system including: (1) research nurses working on the POLARISE trial; (2) POLARISE trial staff and (3) staff involved in the management of POLARISE trial data. Research staff will be enrolled to evaluate the feasibility, acceptability and ease of use of the clinician-facing components of the PROmics system (eg, clinical dashboard) and to explore perceptions and attitudinal responses to ePRO data capture within an ATMP trial.

All PROmics participants must be able to converse in everyday English and be 18 years old or older. Exclusion criteria include being unable to read, understand and complete questionnaires presented in English.

## Recruitment and consent

All POLARISE patient participants will be eligible to take part in this feasibility study and will be provided

with information packs relating to the feasibility study on enrolment in the POLARISE trial. The first patient will be registered to POLARISE on 19 April 2022 and the trial is estimated to recruit for 2 years. POLARISE participants will be involved in the trial for approximately 25 months. The estimated duration of the planned feasibility study is dependent on trial recruitment. Interviews will be scheduled with patient participants 4–6 months after trial enrolment .

The PROmics feasibility study information packs will be tailored depending on whether a POLARISE participant has: (1) consented to report their PRO data via the PROmics system (accept); (2) consented to provide PRO data via the PROmics system and changed to paper-based PROs (withdraw) or (3) declined to report their PRO data via the PROmics system (decline). All versions (accept/withdraw/decline) of the information pack will include a cover letter, participant information sheet and an expression of interest form (+ reply paid envelope) to be returned by the participant to the PROmics research team if they wish to take part in the feasibility study. POLARISE participants who decline to take part in the feasibility study (ie, do not return an expression of interest form) will complete the PROs using their preferred method (ie, using PROmics or completing paper forms) and will have no further contact with the PROmics research team.

POLARISE research team participants will be approached and provided with recruitment packs through liaison with the POLARISE trial sites. The POLARISE research staff recruitment packs will include a cover letter, participant information sheet and expression of interest form.

A semistructured interview conducted virtually (eg, via telephone or videoconference) will be arranged with all participants (patients and members of the POLARISE research team) who express an interest in participating in this feasibility study. Verbal consent will be sought at the time of the telephone/videoconference interview. A prespecified consent script will enable consent to be taken virtually. Participants will be reminded of the study aims and that they are able to withdraw from the study at any time. The consenting process will be completed prior to commencing the interview and will be audiorecorded.

### Data collection

Interviews will be arranged at a time point in the POLARISE trial to allow participants sufficient opportunity to use the PROmics system and to ensure accurate recall of their experiences (ie, approximately 4–6 months after POLARISE trial enrolment). As the POLARISE trial is a national trial with multiple sites, this method of data collection was selected to enable: (1) ease of communication between participants and the PROmics research team and (2) ensure safe social distancing for COVID-19.

Semistructured topic guides (see online supplemental file 1) will support patient and research staff interviews and will ensure key topics are consistently covered. The guides will include set/specific questions designed to elicit descriptions of participants' experiences using the PROmics system, product perception, compliance, acceptability and feasibility of use. The semistructured nature of the planned interviews will enable the researcher conducting the interviews to explore topics in depth and accommodate emerging themes through prompts and questions that are informed by the participants' responses. All interviews will be audiorecorded using a digitally encrypted recorder and transcribed verbatim by a qualified, professional transcription service working within a confidentiality agreement. To confirm the accuracy of the interview transcripts, the transcripts will be reviewed against the audiorecording by the PROmics research team. All transcripts will be deidentified and a unique identifier will be assigned to each participant prior to undertaking data analyses. Data collection and analysis will take place concurrently and will continue until the research team agrees data saturation has been reached (per participant group and overall).

### Data analysis

The deidentified interview transcripts will be entered into NVivo (V.20) qualitative data analysis software and analysed using thematic analysis and the framework method to identify, describe and report patterns in the data.[30–32] The framework method is considered appropriate for use with concrete data in studies where research outcomes are clearly defined.[32] Two researchers from the PROmics research team (CM and SEH) will independently code six interview transcripts (three per participant group). Coding will proceed inductively line-by-line to develop a set of initial codes. The researchers will compare codes to produce a consensus-based coding framework which will be checked by the wider research team before being applied across the data set. Codes will be grouped to form categories and categories refined to represent higher-level themes which encompass the full data set and present a clear, coherent and detailed interpretation of the data. Each participant group (ie, patients and research staff) will be analysed separately and then compared later in the analysis. The final analysis will be discussed with the POLARISE trial Patient and Public Involvement and Engagement (PPIE) group to establish the trustworthiness and validity of the findings.

### Reflexivity

Reflexivity may be defined as researchers' acknowledgement of their own beliefs, judgements and practices and the role these play during the research process.[33] It is central to conducting a thematic analysis as the researcher plays an active role in the identification of patterns within the data.[30 31] The PROmics research team comprises experts in PROs (MC, SEH and CM), qualitative research (CM and SEH), clinical trials (AR and RM), ePROs (CF and EHD) and a consultant hepatologist (PNN) and is nested within the larger POLARISE trial research team. Two experienced qualitative researchers (CM and SEH) will conduct the interviews and undertake data analysis.

Reflexive research journals and memo-writing will be used to acknowledge the role of the researcher lens during the analytical process. None of the research team have any relationship with patient participants; however, it is possible that participants who are research staff at POLARISE trial sites may be known contacts to some members of the PROmics team.

## Patient and public involvement

PPIE in the PROmics study and wider POLARISE trial was informed by the National Institute for Health Research guidance on the involvement of patient and public contributors to research.[34] Patient partners were involved from the outset of the project. To date, PPIE members have informed the design and content of the PROmics ePRO system. A series of stakeholder meetings were convened to establish patient preferences for the format and layout of the PROmics system and to inform PRO selection. Prior to deployment in the POLARISE trial, usability testing of the PROmics system was completed with PPIE members. The outcomes of PROmics stakeholder engagement events and usability testing are reported elsewhere. This protocol was reviewed by patient partner (RW) and the Liver and Gastro-Intestinal PPI Reference Group at the University of Birmingham and the Birmingham Rheumatology Research Patient Partnership (R2P2) who will also support the dissemination of information about the study. Patient and public involvement will be evaluated at the end of the project using the GRIPP-2 checklist.[35 36]

## ETHICS AND DISSEMINATION

Ethical approval for the PROmics feasibility study was granted as part of the POLARISE trial approval by London—West London and GTAC Research Ethics Committee (21/LO/0475). Local R&D approvals will be obtained prior to recruitment at sites. Informed consent will be obtained verbally and audiorecorded from all participants before data collection. Participants will consent to being interviewed and the interviews being audiorecorded and transcribed verbatim. They will acknowledge that their words may be used anonymously in the presentation of the research, that they may withdraw their data within seven working days of completing the interview, that participation is voluntary and that data management and storage is subject to the UK General Data Protection Regulation (2018). The results of this study will be disseminated via peer-reviewed journals, conference presentations and shared via social media. The PPIE groups and ATMP stakeholder networks will be consulted to maximise dissemination and impact.

## DISCUSSION

Trialists, regulators and policy-makers increasingly recognise the importance of the patient experience in the evaluation of the effectiveness, safety and tolerability of medicinal products, including in trials of advanced cellular and gene therapies.[2 37] Accordingly, PRO assessments are included in clinical trials to measure constructs such as symptom severity, functioning, and health-related QoL and, increasingly, these data are collected using ePROs. However, the impact on patients and the feasibility and acceptability of ePRO data capture has yet to be established fully in ATMP trials. This qualitative study aims to provide a deeper understanding into the use ePROs in clinical trials of advanced therapies. Barriers and enablers to the use of PROmics, a new ePRO data capture system, will be explored within the POLARISE trial, a phase II ATMP trial for patients with immune-mediated inflammatory disease. These qualitative data will be supplemented with further quantitative findings at the end of the POLARISE trial including compliance and rates of missing data. The findings will inform recommendations to improve the PROmics system, to enhance PRO strategy in future ATMP trials; promote efficient use of measures, data completeness and acceptability to trial participants; inform future analyses and sample size estimation; and support future implementation of ePROs in ATMP trials and routine care.

The study will incorporate the views of a range of stakeholders. The involvement of trial participants from different disease cohorts and from different geographical locations in the UK will enhance variation in the study sample. These patient participants will provide insights into user confidence with digital technology and the completion of PROs electronically, time points of administration and the overall user experience. Inclusion of trial participants who declined or withdrew from using the PROmics will allow exploration of potential barriers and solutions to participation including inclusion of underserved populations. The views of trial staff and site research nurses will help understand PROmics integration with local IT services and the administrative burden on staff as well as barriers to and enablers of the use of PROmics for data collection, trial participant surveillance and AE reporting.

The use of qualitative methods is a strength of this study, allowing in-depth exploration of user experience. Nesting the feasibility study within the POLARISE trial will enable timely data collection thereby limiting recall bias. The study sample will include research participants from NHS hospitals from different regions of the UK which may enhance the transferability and generalisability of the study findings. However, as not all trial participants will take part in the feasibility study, sampling is potentially biased. Digital exclusion, particularly for underserved populations that may have limited access to digital devices, contributes a further source of bias to the proposed study.[38] Sampling bias will be addressed by recruiting purposively for maximum variation and through the inclusion of participants who declined to use the PROmics system or who reverted to the use of paper-based PROs

at some point in the trial. Questionnaires programmed onto the PROmics system are currently in UK English; however, if the solution proves feasible, translated and culturally validated measures will be included in future work.

The number of ATMP trials continues to grow year-on-year, offering potentially powerful new treatments for patients with rare or life-threatening diseases.[39] As with all investigational medicinal products, demonstrating efficacy, safety and tolerability of new ATMPs is crucial. ePROs have potential to contribute important evidence of efficacy, safety and tolerability (both in the short and longer term) for these novel therapies and to support patient monitoring and risk management. Data collected via the PROmics system could provide new insights into the reporting of symptomatic AEs specific to the advanced therapies context while evaluation of the feasibility and acceptability of the PROmics system will contribute new knowledge to support the development of ePRO reporting formats that meet the unique needs of advanced therapies, for the benefit of all who receive these novel treatments.

**Author affiliations**
[1]Centre for Patient Reported Outcome Research, Institute of Applied Health Research, University of Birmingham, Birmingham, UK
[2]National Institute for Health and Care Research (NIHR) Applied Research Centre (ARC) West Midlands, University of Birmingham, Birmingham, UK
[3]Birmingham Health Partners Centre for Regulatory Science and Innovation, University of Birmingham, Birmingham, UK
[4]National Institute of Health and Care Research (NIHR) Oxford-Birmingham Blood and Transplant Research Unit (BTRU) in Precision Therapeutics, University of Birmingham, Birmingham, UK
[5]National Institute fo Health and Care Research (NIHR) Surgical Reconstruction and Microbiology Research Centre, University of Birmingham, Birmingham, UK
[6]Centre for Trauma Research, University of Birmingham, Birmingham, UK
[7]National Institute for Health and Care Research (NIHR) Birmingham Biomedical Research Centre, University of Birmingham, Birmingham, UK
[8]Cancer Research UK Clinical Trials Unit, University of Birmingham, Birmingham, UK
[9]Aparito, Wrexham, UK
[10]Cognitant, Oxford, UK
[11]Birmingham Law School, University of Birmingham, Birmingham, UK
[12]National Cancer Research Institute (NCRI) Consumer Forum, Sarcoma Patients Euronet, Church Stretton, UK
[13]Institute of Immunology and Immunotherapy, University of Birmingham, Birmingham, UK
[14]Midlands Health Data Research UK, University of Birmingham and Institute of Applied Health Research, Birmingham, UK
[15]DEMAND Hub, University of Birmingham, Birmingham, UK

**Acknowledgements** The authors thank POLARISE Trial Steering Committee, POLARISE Scientific Advisory Group, PROmics Project Management Group, and Patient and Public Involvement (PPI) group members for their input and support of the project.

**Contributors** The feasibility study concept and design were conceived by ARo, ARe, PNN, MC, RW, RH and GP. CB, ARo will support recruitment. EHD, CF and GM were responsible for the development and programming of PROmics in Atom5. EHD and CF will support POLARISE data collection using the PROmics system. SEH and CM will consent participants and undertake the interviews and data analysis. The protocol was written by ARe, RM, MC, ARo, RW and PNN. SEH prepared the first draft of the manuscript. AW will provide administrative support to the project. All authors (SEH, CM, ARo, ARe, RM, CB, EHD, CF, GM, RH, GP, RW, PNN, MC and AW) reviewed the manuscript, provided edits and approved the final version.

**Funding** The author(s) disclosed receipt of the following financial support for the research, authorship and/or publication of this article: This work was supported by the National Institute for Health Research (NIHR) Birmingham Biomedical Research Centre, Midlands-Wales Advanced Therapies Treatment Centre (MW-ATTC) programme grant from Innovate UK to a consortium of partners including Health Technology Wales, the Welsh Blood Service and the University of Birmingham (Grant number: IUK: 104232) and Innovate UK (part of UK Research and Innovation) grant Patient-reported outcomes assessment to support accelerated access to advanced cell and gene therapies: PROmics (Grant No: IUK: 104777).

**Disclaimer** The funder was not involved in any aspect of the research work.

**Competing interests** SEH receives funding from the NIHR Oxford-Birmingham Blood and Transplant Research Unit (BTRU) in Precision Therapeutics, UK Research and Innovation (UKRI) and declares personal fees from Cochlear, Astra Zeneca, and Aparito. MC is a National Institute for Health Research (NIHR) senior investigator and receives funding from the National Institute for Health Research (NIHR) Birmingham Biomedical Research Centre, the NIHR Oxford-Birmingham Blood and Transplant Research Unit (BTRU) in Precision Therapeutics, the NIHR Surgical Reconstruction and Microbiology Research Centre and NIHR ARC West Midlands at the University of Birmingham and University Hospitals Birmingham NHS Foundation Trust, Health Data Research UK, UKRI, Macmillan Cancer Support, UCB Pharma Gilead, Janssen and GSK. MC has received personal fees from Astellas, Aparito, CIS Oncology, Takeda, Merck, Daiichi Sankyo, Glaukos, GSK and the Patient-Centred Outcomes Research Institute (PCORI) outside the submitted work. CM receives funding from NIHR Surgical Reconstruction and Microbiology Research Centre (SRMRC), the NIHR Oxford-Birmingham Blood and Transplant Research Unit (BTRU) in Precision Therapeutics UKRI, and declares personal fees from Aparito outside the submitted work. All other authors have no interests to declare.

**Patient and public involvement** Patients and/or the public were involved in the design, or conduct, or reporting, or dissemination plans of this research. Refer to the Methods section for further details.

**Patient consent for publication** Not applicable.

**Provenance and peer review** Not commissioned; externally peer reviewed.

**ORCID iDs**
Sarah E Hughes http://orcid.org/0000-0001-5656-1198
Christel McMullan http://orcid.org/0000-0002-0878-1513
Ameeta Retzer http://orcid.org/0000-0002-4156-8386
Roger Wilson http://orcid.org/0000-0002-6043-7306
Melanie Calvert http://orcid.org/0000-0002-1856-837X

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
