## [Reviewer comments · BMJ Open]

ARTICLE DETAILS

TITLE (PROVISIONAL)	Feasibility of a new electronic patient-reported outcome (ePRO) system for an advanced therapy clinical trial in immune-mediated inflammatory disease (PROmics): protocol for a qualitative feasibility study
AUTHORS	Hughes, Sarah; McMullan, Christel; Rowe, Anna; Retzer, Ameeta; Malpass, Rebecca; Bathurst, Camilla; Davies, Elin Haf; Frost, Chris; McNamara, Gary; Harding, Rosie; Price, Gary; Wilson, Roger; Walker, Anita; Newsome, Philip; Calvert, Melanie

VERSION 1 – REVIEW

REVIEWER	Andre Scherag Universitätsklinikum Jena
REVIEW RETURNED	09-May-2022

GENERAL COMMENTS	This is a study protocol for an embedded qualitative feasibility study of a new electronic patient-reported outcome systems. The title should indicate that the study is concerned with feasibility and is not a feasibility study itself (which would make little sense). Reading the paper is all fine - however, content (what do you plan to ask?) is lacking and would be of central interest to the readers. So please make sure to include information on content what the structured interview. As two minor remarks: a) Please also highlight which part of the interviews is in a multicenter setting and which his not. b) what if the patients who declined or withdrew also have no interest in the interview?
---

REVIEWER	Hollie Richards University of Bristol, Population Health Sciences
REVIEW RETURNED	10-May-2022

GENERAL COMMENTS	Thanks for the invitation to review this interesting and important protocol paper for a qualitative sub-study within the POLARISE Trial to evaluate the patient and researcher experiences of using an ePRO system to report PROMs relating to ATMPs. I have a few minor comments and requests for clarification below: 1. Recruitment and methods: - There is an inconsistency in the planned number of participants (10-15 in the abstract and 20-30 in the methods section) - At what point in recruitment to the POLARISE Trial will participants be recruited to this qualitative study? Could you include some brief details (e.g. at the same time as recruitment to POLARISE or at any other convenient timepoint after they have already been enrolled on the main trial?) - It would be good to have a little more detail about how participants
--

	from the wider POLARISE Trial will be identified - will there be a selection process to identify trial participants who fulfil eligibility criteria 1? Although the number of participants you plan to interview seems appropriate for this study it's unclear how criteria 1 interview participants will be identified from the much larger trial cohort. 2. Apologies if I have missed this but it would be good to have an idea of the planned number of sites for the wider trial, and therefore this qualitative study. 3. It would be useful to know more about the PROmics system - is it a commercially available system or has it been developed for this trial?
--	--

REVIEWER	Elizabeth Papautsky Patient reviewer, Department of Biomedical & Health Information Sciences
REVIEW RETURNED	12-May-2022

GENERAL COMMENTS	The authors describe a protocol for qualitative feasibility study of a new electronic patient-reported outcome (ePRO) system. This is a topic that is of much needed attention. PROs are not defined and explained sufficiently, and concrete examples are missing. PROs range from objective data from e.g. fitbit to qualitative symptom reporting – this doesn't come through in the current narrative. Also, need to be explicit about the value and benefits of PROs – to clinicians and to patients and to both. The section 'PROs in trials of ATMPs' covers some of this, but more detail and justification is needed. In addition, PROs have the potential to contribute to workload for clinicians – what's the argument for the benefit outweighing the cost? Page 6, Line 23: need references and examples In Table 1, the rationale for selection is appreciated, but there also needs to be justification for why this information from the patient is important. Concern that too many disease types (4) are chosen. Consider focusing on one to start with. Authors proposing remote collection of PROs, which is fine, but need to demonstrate that this type of data collected will not yield a representative sample as underserved populations may have limited access to technology. Page 8, Line 52: need more detail for the 'Design' section Page 8/9, Line 60: The following statement needs to be edited/reworded: "...defined as the degree to which new data express what was expressed in previous data..." Need justification for the inclusion of research staff in the study sample. Authors bring up 'Semi-structured topic guides' and some general topics of interest, but stop there. There needs to be more specific details describing the interviews. At the least, please include some specific questions as examples, if not include a full (draft) interview guide. Otherwise, the reader is unable to fully understand the knowledge that you propose to elicit from participants. The data analysis section is generic and needs to include specifics around the topic at hand.
--

REVIEWER	Rosie Cornish University of Bristol, School of Social and Community Medicine
REVIEW RETURNED	14-May-2022

GENERAL COMMENTS	This is a clearly written and comprehensive protocol. I have only a couple of minor comments. Minor comments 1. At the end of the participant selection criteria the authors list exclusion criteria as being declining the optional consent for the use of the PROmics system. However, in the next section (recruitment and consent) it says that these individuals will be sent an information pack which includes an expression of interest form – is this in case they change their mind and decide that they are happy to use the PROmics system? 2. I noted a few minor typos: (a) Abstract – line 12 on page 4 of 14: “10-15 patients in enrolled in”; (b) line 3 or 4 on page 8 of 14 (page 6 of manuscript): “are delivered in a standardised ordered” (should be order?); (c) line 54 on page 9 of 14 (page 7 of manuscript): “prior to commencing the interview and audio recorded” (should be recording?); (d) line 3 on page 12 of 14 (page 10 of manuscript): ...”to grow year-on-year; offering” (should be a comma here not a semi-colon).
---

VERSION 1 – AUTHOR RESPONSE

Reviewer 1:

This is a study protocol for an embedded qualitative feasibility study of a new electronic patient-reported outcome systems. The title should indicate that the study is concerned with feasibility and is not a feasibility study itself (which would make little sense).

Thank you, we have changed the title to reflect the proposed changes suggested by the editor (page 1, lines 1-4).

Reading the paper is all fine - however, content (what do you plan to ask?) is lacking and would be of central interest to the readers. So please make sure to include information on content what the structured interview.

Thank you we have include the interview topic guide as a supplemental file.

As two minor remarks: a) Please also highlight which part of the interviews is in a multicenter setting and which his not. b) what if the patients who declined or withdrew also have no interest in the interview?

- a) All of the interviews will be conducted remotely via telephone or videoconferencing. The POLARISE trial is a multi-centre trial and the PROmics team conducting this feasibility study will invite all participants who enrol in the trial (+trial research staff), see page 8, line 284-285.
- b) Participants who decline the PROmics system may also decline to take part in the feasibility study interview. Participants who decline to use the PROmics system and decline an interview may do so without this affecting their involvement in the POLARISE trial. All PROs will be collected using paper forms and the PROmics team will have no further contact with the trial participant. We have added the following text (page 7, lines 275-278):

“POLARISE participants who decline to take part in the feasibility study (i.e., do not return an expression of interest form) will complete the PROs using their preferred method (i.e., using PROmics or completing paper forms) and will have no further contact with the PROmics research team.”

Reviewer 2:

1. Recruitment and methods:

- **There is an inconsistency in the planned number of participants (10-15 in the abstract and 20-30 in the methods section).**

Thank you, the abstract stated 10-15 patients and 10-15 research team members (N = 20-30). We have revised the abstract for clarity (page 2, lines: 58-61):

“10-15 patients enrolled in the POLARISE trial, a single arm, phase II ATMP basket trial, and 10-15 research team members at the ATMP trial sites will be recruited.”

- **At what point in recruitment to the POLARISE Trial will participants be recruited to this qualitative study? Could you include some brief details (e.g. at the same time as recruitment to POLARISE or at any other convenient timepoint after they have already been enrolled on the main trial?)**

The manuscript reads as follows: “Interviews will be arranged at a time point in the POLARISE trial to allow participants sufficient opportunity to use the PROmics system and to ensure accurate recall of their experiences whilst enrolled in the POLARISE trial”. For clarity we have added “(i.e., approximately 4-6 months after POLARISE trial enrolment)”. (page 8, line 296)

- **It would be good to have a little more detail about how participants from the wider POLARISE Trial will be identified - will there be a selection process to identify trial participants who fulfil eligibility criteria 1? Although the number of participants you plan to interview seems appropriate for this study it's unclear how criteria 1 interview participants will be identified from the much larger trial cohort.**

The POLARISE trials aims to recruit 60 patients. All POLARISE participants will be invited to take part in the PROMics feasibility study at the time of trial enrolment as stated in the manuscript (page 7, line 262):

“All POLARISE patient participants will be eligible to take part in this feasibility study”

For clarity we have added a study flow diagram to the manuscript (Figure 1).

2. Apologies if I have missed this but it would be good to have an idea of the planned number of sites for the wider trial, and therefore this qualitative study.

We have added this detail to the manuscript under “study design” (page 7, lines 222-224): “A remote, qualitative, interview-based feasibility study (see figure 1) embedded within the POLARISE trial, a single arm, phase II ATMP basket trial recruiting from six National Health Service (NHS) trusts in the United Kingdom.

3. It would be useful to know more about the PROMics system - is it a commercially available system or has it been developed for this trial?

Thank you for your comment. The PROMics system is a bespoke, ePRO system developed specifically for the POLARISE trial. We have amended the manuscript as follows:

“The PROMics system is a bespoke, trial-specific electronic data capture system designed to collect and assess PRO data when patients receive an advanced cell therapy.” (page 4, lines 145-146)

Reviewer 3:

PROs are not defined and explained sufficiently, and concrete examples are missing. PROs range from objective data from e.g. fitbit to qualitative symptom reporting – this doesn't come through in the current narrative.

Thank you for your feedback. PROs are defined in the manuscript using the FDA 2009 definition. We suggest that wearables are objective measures of patient activity that complement PROs but are distinct from PROs. We have deliberately not included examples as these may be perceived as endorsement.

We have added the following to highlight the use of wearables alongside PROs (page 4, lines 109-112):

“PRO data include patient perspectives of symptom burden, functioning, and their health-related quality of life and are typically captured using validated, self-report questionnaires. PROs may be used alone or in combination with wearable devices to collect real-time, objective measures of patient activity.”

Also, need to be explicit about the value and benefits of PROs – to clinicians and to patients and to both. The section ‘PROs in trials of ATMPs’ covers some of this, but more detail and justification is needed.

We have limited our discussion to the value and benefits of PROs in clinical trials, adding the following to ensure the patient and clinician view is included (page 4, lines 135-142):

“In addition, PROs have potential to support a fuller understanding of tolerability (the ability or desire of the patient to adhere to a specific dose or intensity of therapy) by providing direct measurements from the patient on how they are feeling and functioning while on treatment.^{13 14} For patients, PROs in ATMP trials provide an opportunity to communicate outcomes of importance not captured by traditional clinical endpoints. PROs may also encourage patients to engage as participants, increase the likelihood of PRO claims in product labelling, and empower patients and clinicians to make more informed treatment decisions leading to better clinical outcomes.¹⁵”

In addition, PROs have the potential to contribute to workload for clinicians – what’s the argument for the benefit outweighing the cost?

Thank you for your comment. We agree that PROs are associated with both costs and benefits for clinicians, especially regarding workload. However, this protocol focusses specifically on the use of PROs in an early-phase trial setting. We politely disagree with the need to include a discussion relating to the use of PROs in routine care on the basis that a discussion with this focus sits outside the manuscript’s narrative.

Page 6, Line 23: need references and examples

We were unsure what the reviewer is requesting. We have added the following reference at page 6, line 194.

David T, Ling SF, Barton A. Genetics of immune-mediated inflammatory diseases. *Clin Exp Immunol* 2018;193:3–12. doi:10.1111/cei.13101

In Table 1, the rationale for selection is appreciated, but there also needs to be justification for why this information from the patient is important.

Thank you for your feedback we have added the following to the body of the manuscript (page 5, lines 150-152):

“The included PROs were selected to facilitate assessment of efficacy and tolerability by gathering preliminary evidence from the patient perspective on the benefits and risks of the ATMP including therapeutic responses, tolerability and toxicity. These data may also inform the design and conduct of later-phase trials.”

Concern that too many disease types (4) are chosen. Consider focusing on one to start with.

Thank you for your comment. We are keen to present a representative account of the POLARISE trial participants' experiences using the PROMics system. As these may differ across disease groups, we opted to include all four groups in our study sample; however, we will look both within and across groups when undertaking our analysis (page 9, lines 329-330):

“Each participant group (i.e., patients and research staff) will be analysed separately and then compared later in the analysis.”

Authors proposing remote collection of PROs, which is fine, but need to demonstrate that this type of data collected will not yield a representative sample as underserved populations may have limited access to technology.

We have included mention of digital exclusion as a possible source of sampling bias in the strengths and limitations section of the manuscript. We have added the following (page 10, lines 409-410):

“Digital exclusion, particularly for underserved populations that may have limited access to digital devices, contributes a further source of bias in the proposed study.”

Page 8, Line 52: need more detail for the 'Design' section.

We have refined this section which now reads as follows (page 7, lines 222-224):

“A remote, qualitative, interview-based feasibility study (see figure 1) embedded within the POLARISE trial, a single arm, phase II ATMP basket trial recruiting from six National Health Service (NHS) trusts in the United Kingdom.”

Page 8/9, Line 60: The following statement needs to be edited/reworded: “...defined as the degree to which new data express what was expressed in previous data...”

Thank you, for highlighting this typo which has now been fixed in the manuscript (page 7, lines 203-232).

“...defined as the degree to which new data expresses what was expressed in previous data...”

Need justification for the inclusion of research staff in the study sample.

Thank you for your comments. We have included the following (page 7, lines 253-255):

“Research staff were enrolled to evaluate the feasibility, acceptability and ease of use of the clinician-facing components of the PROmics system (e.g., clinical dashboard) and to explore perceptions and attitudinal responses to ePRO data capture within an ATMP trial.”

Authors bring up ‘Semi-structured topic guides’ and some general topics of interest, but stop there. There needs to be more specific details describing the interviews. At the least, please include some specific questions as examples, if not include a full (draft) interview guide. Otherwise, the reader is unable to fully understand the knowledge that you propose to elicit from participants.

Thank you for your comment. We have included the interview topic guides as a supplemental file and highlight the following sentence in the manuscript (page 8, lines 301-304):

“Semi-structured topic guides (see online supplemental file 1) will support patient and research staff interviews and will ensure key topics are consistently covered. The guides will include set/specific questions designed to elicit descriptions of participants’ user experiences of the PROmics system, product perception, compliance, acceptability, and feasibility of use.”

The data analysis section is generic and needs to include specifics around the topic at hand.

We have expanded the “Data analysis” and “Reflexivity” section of the manuscript, adding detail to the description of the planned data analysis (pages 8-9, lines 317-344).

Reviewer 4:

- 1. At the end of the participant selection criteria the authors list exclusion criteria as being declining the optional consent for the use of the PROmics system. However, in the next section (recruitment and consent) it says that these individuals will be sent an information pack which includes an expression of interest form – is this in case they change their mind and decide that they are happy to use the PROmics system?**

Thank for your comments. All POLARISE participants irrespective of their status (accept/withdraw/decline PROMics) will have the opportunity to participate in the feasibility study. Patients who have “declined” to use PROMics may still wish to take part in the feasibility study. Indeed, interviews with POLARISE participants who decline to use the PROMics system will be an important element of the feasibility study, informing understanding of barriers to uptake. We have refined the manuscript for greater clarity (pages 7-8, lines 261-278) and added a flow diagram (Figure 1).

2. I noted a few minor typos: (a) Abstract – line 12 on page 4 of 14: “10-15 patients in enrolled in”; (b) line 3 or 4 on page 8 of 14 (page 6 of manuscript): “are delivered in a standardised ordered” (should be order?); (c) line 54 on page 9 of 14 (page 7 of manuscript): “prior to commencing the interview and audio recorded” (should be recording?); (d) line 3 on page 12 of 14 (page 10 of manuscript): ...”to grow year-on-year; offering” (should be a comma here not a semi-colon).

Thank you, these typos have now been corrected.

VERSION 2 – REVIEW

REVIEWER	Hollie Richards University of Bristol, Population Health Sciences
REVIEW RETURNED	25-Jul-2022

GENERAL COMMENTS	Thanks for addressing my comments - the recruitment flow diagram makes it much clearer.
---

REVIEWER	Elizabeth Papautsky Patient reviewer, Department of Biomedical & Health Information Sciences
REVIEW RETURNED	04-Aug-2022

GENERAL COMMENTS	The manuscript is much improved with the authors’ edits in response to reviewer comments. I have only minor comments below:  - The first sentence in the ‘Aims and Objectives’ section is incomplete. - This is the same comment that I previously made and it was not addressed – the following is awkward wording and needs to be edited: “degree to which new data expresses what was expressed in previous data” - Page 7, line 255: needs to be future tense - Page 8, line 291: “the telephone” should be changed to “virtually” (or something similar) since the interviews may be conducted via telephone or via video conferencing
--

VERSION 2 – AUTHOR RESPONSE

***There seems to have been some confusion regarding one of our previous requests. We asked you to change the heading ‘STRENGTHS AND LIMITATIONS’ to ‘Strengths and limitations of this study’. This referred to the heading of the section after the abstract containing bullet points. Instead, the authors have added headings ‘Strengths and limitations of this study’ and**

'Conclusions' to paragraphs in the Discussion section. This is not appropriate, as we do not allow Conclusions sections in protocol manuscripts. Please remove these added subheadings and instead change the heading of the section after the abstract, as requested.

Please accept our apologies for this misunderstanding. We have amended the manuscript to include the heading "Strengths and limitations of this study" and remove the subheadings in the discussion (page 10, lines 389 and 403).

***Thank you for adding the information on the study setting to the 'Design' section of the main text. Please also update the ABSTRACT 'Methods and analysis' section to include the same information on the study setting.**

We have made the suggested change, adding the following text:

"This protocol is for a remote, qualitative, interview-based feasibility study embedded within the POLARISE trial, a single arm, phase II, multi-site ATMP basket trial in the United Kingdom. 10-15 patients enrolled in the POLARISE trial and 10-15 research team members at the ATMP trial sites will be recruited." (page 2, lines 55-58).

***Thank you for adding detailed information on participant consent procedures to the 'ETHICS AND DISSEMINATION' section of the main text. Please also update the ABSTRACT 'Ethics and dissemination' section to include a very brief sentence on participant informed consent requirements.**

We have added the following sentence: "Informed consent will be obtained from all participants prior to data collection." (page 2, line 63).

- The first sentence in the 'Aims and Objectives' section is incomplete.

We have amended the text as follows:

This study aims to qualitatively assess the feasibility and acceptability of the PROmics system, an ePRO platform for use in trials of ATMPs." (page 6, line 197)

- This is the same comment that I previously made and it was not addressed – the following is

awkward wording and needs to be edited: “degree to which new data expresses what was expressed in previous data”

Apologies for this omission. We have amended the manuscript as follows:

“Recruitment will proceed until data saturation, the point in data collection when new data are redundant, is achieved.[29]”

- Page 7, line 255: needs to be future tense

We have made this change as suggested.

- Page 8, line 291: “the telephone” should be changed to “virtually” (or something similar) since the interviews may be conducted via telephone or via video conferencing

Thank you, we have made this amendment and the manuscript reads as follows:

“A semi-structured interview conducted virtually (e.g., via telephone or videoconference)...” (page8, line 275)